# Ductular Reactions in Liver Injury, Regeneration, and Disease Progression—An Overview

**DOI:** 10.3390/cells13070579

**Published:** 2024-03-26

**Authors:** Nirmala Mavila, Mallikarjuna Siraganahalli Eshwaraiah, Jaquelene Kennedy

**Affiliations:** 1Karsh Division of Gastroenterology and Hepatology, Department of Medicine, Cedars-Sinai Medical Center, Los Angeles, CA 90048, USA; mallikarjuna.siraganahalli@cshs.org (M.S.E.); jaquelene.kennedy@cshs.org (J.K.); 2Division of Applied Cell Biology and Physiology, Department of Biomedical Sciences, Cedars-Sinai Medical Center, Los Angeles, CA 90048, USA

**Keywords:** ductular reaction, liver injury, liver fibrosis, liver progenitors

## Abstract

Ductular reaction (DR) is a complex cellular response that occurs in the liver during chronic injuries. DR mainly consists of hyper-proliferative or reactive cholangiocytes and, to a lesser extent, de-differentiated hepatocytes and liver progenitors presenting a close spatial interaction with periportal mesenchyme and immune cells. The underlying pathology of DRs leads to extensive tissue remodeling in chronic liver diseases. DR initiates as a tissue-regeneration mechanism in the liver; however, its close association with progressive fibrosis and inflammation in many chronic liver diseases makes it a more complicated pathological response than a simple regenerative process. An in-depth understanding of the cellular physiology of DRs and their contribution to tissue repair, inflammation, and progressive fibrosis can help scientists develop cell-type specific targeted therapies to manage liver fibrosis and chronic liver diseases effectively.

## 1. Introduction

Out of all the vital organs, the liver has the best ability to repair itself following an injury. Under normal conditions, hepatic cell proliferation occurs at the minimum levels. During acute injury, the replication of liver cells occurs via precisely regulated and complicated cell signaling mechanisms to attain tissue homeostasis [1,2,3]. An acute hepatic injury triggers hepatocyte proliferation, while damage to the bile ducts induces cholangiocyte expansion (Figure 1). Paracrine factors released from hepatic stellate and endothelial cells play a crucial role in liver regeneration [3,4]. During chronic bile duct injury, as in cholangiopathies, cholangiocyte proliferation compensates for the cellular loss. This eventually progresses to a more complex cellular response, often described as ductular reactions (DRs) [5]. A DR is defined as a complex heterogeneous cellular response that occurs during chronic liver injury, consisting of mainly cholangiocytes derived from reactive cholangiocytes and, to a lesser extent, cells originating from liver progenitors (LPCs) and transdifferentiated hepatocytes [6]. Over time, a DR induces extensive morphological remodeling in the liver with the additional involvement of portal mesenchyme and infiltrated immune cells. DRs develop in many chronic liver diseases with different etiologies, where they show a strong correlation with the disease severity [7]. Cholangiopathies such as biliary atresia (BA), primary sclerosing cholangitis (PSC), primary biliary cirrhosis (PBC), and other chronic conditions such as alcoholic hepatitis (AH), nonalcoholic steatohepatitis (NASH), and viral hepatitis are some of the liver diseases associated with DRs [7,8,9,10,11,12,13,14,15,16].

The scientific knowledge of liver injury and regeneration mechanisms has increased exponentially in recent decades. However, the role of DRs in the liver regeneration process and how they influence other cell types, such as hepatocytes, hepatic stellate cells (HSCs), immune cells, portal fibroblasts, and endothelial cells in chronic liver diseases, remains unclear. Commonly conserved signaling pathways essential for liver development and regeneration are often dysregulated in disease conditions and have been shown to promote fibrosis and malignant transformation in many experimental models. This review summarizes our fundamental understanding of DRs, explicitly focusing on cellular heterogenicity, molecular drivers that regulate DRs, and their translational significance in chronic liver diseases. 

## 2. Origin and Cellular Heterogeneity in DRs

The cellular heterogenicity and injury-induced reprogramming of liver cells are the two key factors contributing to the formation of DRs. The heterogeneity primarily depends on the nature of an injury and is also attributed to the cellular origin, such as cholangiocytes, hepatocyte-derived cholangiocytes, or LPCs [6].

### 2.1. Cholangiocytes

Cholangiocytes are polarized epithelial cells that line the biliary system, consisting of intra- and extrahepatic bile ducts. Cholangiocytes are generally quiescent and create a safe barrier that protects other cells from toxic bile acids. Moreover, cholangiocytes are heterogeneous phenotypically and functionally, and two distinct populations of cholangiocytes exist; small cholangiocytes line smaller intrahepatic bile ducts and have a higher nucleus-to-cytoplasm ratio and cuboidal morphology with microvilli extending towards the bile duct lumen, and large cholangiocytes, which line larger bile ducts, are columnar and have a small nucleus-to-cytoplasm ratio, with a distinct cilium on the apical surface. In addition to the morphological differences, many functionally essential proteins, including ion channels and membrane receptors, are expressed differently in small and large cholangiocytes. Functionally, larger cholangiocytes regulate bile secretion and its homeostasis and are more susceptible to injury, while smaller cholangiocytes have greater cellular plasticity, are more injury-resistant, and can transform into larger cholangiocytes when these cells are significantly compromised in chronic diseases [17,18,19,20,21]. 

Even though cholangiocytes comprise approximately 3–5% of the liver’s total cell mass, they are essential for normal liver function and serve as a protective cellular barrier. Cholangiocytes participate in the micelle formation and transportation of bile and play a crucial role in maintaining physiological bile homeostasis [22]. This is achieved by specific membrane transporters and exchangers expressed on the apical or basolateral membranes of cholangiocytes. The major transporters and exchangers involved in this process include water channel molecules such as aquaporins, Na+/glucose transporters, and Cl^−^/HCO_3_^−^ exchangers. Additionally, cholangiocytes express xenobiotic enzymes, such as CYP2E1 and CYP1A2 [23,24]. Cholangiocytes secrete factors such as mucins, defensins, and immunoglobulins, such as IgA, thus protecting the biliary system [25]. Membrane receptors such as the cystic fibrosis transmembrane conductance regulator (CFTR), Cl^−^/HCO_3_^−^ anion exchanger 2 (AE2), secretin receptor (SR), somatostatin receptor (SSTR), melatonin receptors (MT1 and MT2), and apical sodium-dependent bile acid transporter (ASBT) expressed on large cholangiocytes regulate bile flow, while the SR axis regulates bicarbonate (HCO_3_^−^) secretion [22]. The apical cell membranes of cholangiocytes possess a single primary cilium that regulates several biological activities, such as cell differentiation, proliferation, and secretion. Smaller cholangiocytes have shorter cilia compared to larger cholangiocytes. Defects in cilia structure and function result in cholangiocyte hyperproliferation and alterations in bile fluidity, thereby affecting bile homeostasis. The loss of primary cilia in cases of cholangiocarcinoma indicates that ciliary proteins may act as tumor suppressors [7,26,27].

Many factors, such as genetic, immunologic, infectious, and obstructive factors, can cause cholangiocyte injury and induce apoptosis, which triggers the activation and proliferation of existing cholangiocytes. Injured and activated cholangiocytes additionally serve as a source of intracellular damage-associated molecular pattern (DAMP) biomolecules, which are released into the extracellular space and initiate and perpetuate a cascade of innate immune and microenvironmental changes that affect other cell types. The factors released from the affected cholangiocytes include soluble molecules and membrane-derived nanometer-sized vesicles, termed extracellular vesicles (EVs), which are known to play a significant role in liver pathobiology [28,29,30,31,32]. Studies have shown that cholangiocytes respond in different ways to various injury conditions. For example, larger cholangiocytes are more susceptible to injury and apoptosis than small cholangiocytes [33]. In a bile duct ligation (BDL) mouse model of liver injury and fibrosis, it was observed that DRs were mainly caused by the proliferation of large cholangiocytes. In contrast, in a carbon tetrachloride (CCl4) model of liver injury and fibrosis, smaller cholangiocytes proliferated and contributed to the formation of DRs [20,34,35]. On the other hand, both small and large cholangiocytes underwent proliferation in bile-acid-treated mice [36]. Cell lineage tracing using transgenic mice demonstrated that cholangiocytes did not proliferate in a uniform manner during injury [37]. This study demonstrated significant cellular heterogeneity in the DR phenotype with the expansion of single isolated cholangiocytes, small bile ductules, and big bile ducts. The differences in both the morphology and how cholangiocytes respond to injuries indicated that small cholangiocytes are more of a conserved and primitive pool of the cell population that resides in the liver, whereas large cholangiocytes are functionally differentiated cells.

### 2.2. Hepatocyte-Derived Cholangiocytes 

Hepatocytes are the primary parenchymal cells in the liver that perform many vital functions. During chronic injury, hepatocytes undergo reprogramming as a reparative mechanism, which is often associated with phenotypic changes and the acquisition of cholangiocyte-like cells [38]. The hepatocyte-derived cholangiocytes are referred to as atypical cholangiocytes and share many characteristics with normal cholangiocytes. Occurrences of hepatocyte-derived cholangiocytes have also been reported in many experimental models of chronic liver injury and disease [39,40,41]. Cell-type-specific immunostaining on liver biopsies using a cholangiocyte marker, cytokeratin-19 (CK19), and a hepatocyte marker, HepPar1, also revealed heterogeneous cell populations in human liver diseases [42,43]. Several other studies have also demonstrated the presence of hybrid or transitional cells that express the markers of cholangiocytes (SOX9, EpCAM) and hepatocytes (HNF4 alpha, HepPar1) [44,45,46,47,48,49,50]. Studies utilizing chimeric animal models pre-treated with biliary toxin methylene diamine (DAPM) and then subjected to bile duct ligation (BDL) revealed the presence of chimeric cholangiocytes with the donor hepatocyte marker dipeptidyl peptidase IV (DPPIV)-positive cells 30 days after BDL surgery, which was further increased 36-fold by the DAPM pretreatment [44]. It was found that SOX9^+^ periportal hepatocytes were reprogrammed into cholangiocyte cell types after a 3,5-diethoxycarbonyl-1,4-dihydrocollidine (DDC)-induced liver injury, were incorporated into damaged bile ducts, and contributed to the DRs [48]. Moreover, Chimeric cholangiocyte cells were developed in vivo by transplanting ROSA26R-mTmG hepatocytes into Fah−/− mice and subjecting them to a partial hepatectomy or cholestatic injury [51]. In a hepatocyte lineage-tracing model using Mx1-Cre-ROSA26R mice, beta-Gal-labeled hepatocytes constituted approximately 1.9% of the cholangiocytes and DRs [52]. A short-term cell lineage study conducted in a zebrafish model demonstrated that hepatocytes acquire the cholangiocyte phenotype without any intermediate states [53].

### 2.3. Liver Progenitor Cells

Even though the liver has the built-in capacity to regenerate itself following an injury, severe and persistent liver injuries can induce cellular senescence, resulting in the loss of the regenerative potential of liver cells. During these circumstances, liver progenitor cells (LPCs) are activated and expand from the canals of Hering (CoH), the terminal branches of the intrahepatic biliary ducts lined with small cholangiocytes and hepatocytes [6]. LPCs are generally not present in a healthy adult liver. A histology analysis using specific molecular markers demonstrated the presence of LPCs in the livers of both human subjects and preclinical animal models of chronic liver diseases. Moreover, the label retention assay revealed the existence of stem cell niches in the CoH [54]. It is hypothesized that signals from the microenvironment control the quiescent nature, stemness, proliferation, and differentiation of LPCs in the CoH [55,56,57]. Several in vivo studies using animal models and human biopsy samples have revealed the expansion of LPCs and their association with DRs [58,59]. Notably, a three-dimensional analysis of necrotic human liver tissues revealed that DRs consist of LPCs originating from the CoH [56].

The activation and differentiation of LPCs are controversial topics of debate concerning their status as de-differentiated mature liver cells [60,61,62]. However, extensive lineage studies utilizing transgenic animal models show strong evidence for the presence of facultative stem/progenitor cells in the CoH and their differentiation potential into both hepatocytes and cholangiocytes [63]. In an animal model of BA, PROM1^+^ progenitors were found to be positive for cholangiocyte markers and profibrogenic markers of fibrosis [11]. Transitional cells that express markers of both progenitors and cholangiocytes are more abundant than intermediate hepatocytes, suggesting that, in biliary diseases, LPCs mainly differentiate towards the cholangiocyte lineage [64,65]. An analysis of human liver samples revealed that cells expressing CD34 could differentiate into cholangiocytes and represented human biliary epithelial progenitor cells [66]. Another recent study demonstrated that CD24^+^LCN2^+^ LPCs in DRs contributed to inflammation in chronic liver damage cases, demonstrating the pathological role of injury-induced LPCs in the liver. This study further demonstrated that CD24^+^LCN2^+^ LPCs enhanced the tissue infiltration of neutrophils and F4/80^+^ macrophages, and silencing *Lcn2* in this cell population eliminated the chemotactic paracrine action on macrophages and prevented the expression of M1 macrophage markers [67]. Apart from the LPCs originating from the CoH, biliary ducts contain specialized glandular structures lined with biliary epithelial cells connected to the bile duct lumen. These are called peribiliary glands (PBGs). Originally, PBGs were identified as the structures that secrete mucinous components into the bile; however, more recent studies have identified the presence of multipotent progenitors, suggesting a secondary pool of LPCs during injury [68,69,70]. In an animal model of BA, PROM1^+^ progenitor cells in the PBG glands are involved in the extrahepatic bile duct regeneration process, suggesting the regenerative potential of LPCs originating from these sites [71].

## 3. Types of Ductular Reactions

In many chronic liver diseases, DRs show differences both phenotypically and morphologically. Based on histopathological features, Desmit classified DRs into four different types [5]. Type 1 DRs are associated with the multiplication of pre-existing cholangiocytes with minimal reorganization of the ductal structure; they maintain an anatomically close relationship with the portal vein just as in the normal liver; and they are reversible. This type of DRs does not establish any new canaliculi–ductular connections. Type 2A DRs occur in periportal areas, as is the case for chronic cholestatic liver diseases, with the involvement of LPCs and cells derived from reprogrammed hepatocytes. Type 2B DRs occur in central–lobular necrotic areas and are induced by hypoxia. Type 3 DRs are associated with LPCs and reprogrammed hepatocytes, and are triggered by extensive hepatic cell death. Type 2A, Type 2B, and Type 3 DRs establish canaliculi–ductular connections and, thus, potentially participate in bile drainage and resolve hepatocyte damage in chronic liver diseases [5]. To simplify this, recent classifications are mainly based on the morphological characteristics of the DRs. A noninvasive DR is also known as a Type 1 or typical DR, and an invasive DR is referred to as a Type 2 /Type 3 or atypical DR. Noninvasive DRs are primarily observed around the periportal area. On the other hand, invasive DRs are characterized by the extensive proliferation of ductules that deeply invade the lobular parenchyma and are accompanied by LPC activation and proliferation. Moreover, invasive DRs are associated with massive hepatic necrosis, parenchymal loss, ductular hyperplasia, LPC expansion, and cellular transformation [72]. Invasive DRs lead to the formation of disorganized tubular structures with poorly defined ductular lumens, which affect the tissue architecture [72]. In humans, invasive DRs were observed in patients with chronic cholestatic liver diseases, nonalcoholic fatty liver disease, alcoholic liver disease, and viral hepatitis C [73,74,75,76,77,78] and in rodent models that were subjected to a choline-deficient diet or thioacetamide administration [79,80,81,82]. In metabolic syndrome, DRs are induced by systemic inflammation triggered by senescent hepatocytes, whereas in hepatitis C infections, insulin resistance and inflammation are the key factors that drive the induction of DRs [77]. Figure 2 summarizes the cellular reprogramming and tissue remodeling events associated with different types of DR development. Invasive DRs associated with cholestatic liver injury and fibrosis comprise reactive cholangiocytes, LPCs, and periductular SOX9^+^ hepatocytes (Figure 2B). In the case of invasive DRs associated with extensive hepatic necrosis, apart from reactive cholangiocytes, LPCs, and periportal SOX9^+^ hepatocytes, transdifferentiated mature hepatocytes are also involved in the formation DRs (Figure 2C). Invasive DRs are also closely associated with extensive inflammation and fibrosis.

## 4. Experimental Liver Disease Models Associated with DRs

Several animal models, mainly using rodents, have been generated to mimic various chronic liver diseases with comparable histopathological characteristics, including DRs. These models represent excellent in vivo tools for investigating the mechanisms of liver injury, regeneration, DRs, LPC activation, fibrosis, cirrhosis, and liver cancer development, and also to test the therapeutic effects of novel drug candidates.
cells-13-00579-t001_Table 1Table 1Commonly used liver disease models associated with DR.Experimental ModelRelevant Human Disease PhenotypeReferencesThioacetamide (TAA) Liver fibrosis, cirrhosis, HCC, CCA[81]Bile duct ligationObstructive cholestasis, fibrosis[83,84]3,5-Diethoxycarbonyl-1,4-dihydrocollidine-diet-induced (DDC) dietPrimary sclerosing cholangitis[85,86]CCl4 injection Fibrosis, cirrhosis[87]Mdr2 knockout micePrimary sclerosing cholangitis[88,89]Rhesus rotavirus (RRV) infectionBiliary atresia[90,91]Diethyl nitrosamine (DEN)Liver fibrosis, cirrhosis, HCC[92]Methionine–choline-deficient (MCD) dietSteatohepatitis[82,93]Choline-deficient, ethionine-supplemented (CDE) diet Steatohepatitis[94,95]

Studies have demonstrated a strong correlation between DRs and disease progression in all these experimental models. Table 1 summarizes the commonly used animal models and their comparable human disease conditions. Detailed information on these models has been reviewed extensively elsewhere [96,97,98,99,100,101,102].

## 5. Molecular Regulation of DRs

Hepatocytes and cholangiocytes are quiescent in the normal liver. However, an injury or insult to the liver induces their replication to repair the damage. A DR consists of a heterogeneous population of hyperproliferative cells, and the extent of cell proliferation depends on the underlying pathology/etiology. Several signaling pathways are known to impact the proliferation of cholangiocytes, both in vivo and in vitro. Injured cholangiocytes secrete various cytokines, growth factors, neuropeptides, and hormones, which play an important role in cell–cell communication and DR formation during chronic injury [103,104,105]. The growth factors released from the inflammatory and stromal cells further induce the proliferation of the biliary epithelium [106]. Cholangiocytes secrete the neuroendocrine hormone serotonin and inhibit the growth of cholangiocytes, both in vitro and in vivo [107]. Furthermore, gain-of-function and loss-of-function approaches have demonstrated the critical role of serotonin signaling in developing DRs and fibrosis in a cholestatic liver injury model in vivo [108]. 

There is evidence of distinct gene expression profiles and different types of signaling pathway activation in small and large cholangiocytes. In vivo studies have demonstrated that H19, a profibrogenic long noncoding RNA, is markedly induced by bile acids in small cholangiocytes compared to large cholangiocytes [109,110]. Secretin, the gastrointestinal peptide hormone, elevates the levels of intracellular cAMP when it interacts with secretin receptors (SRs) expressed on large cholangiocytes, initiating cell proliferation through the protein kinase A (PKA)/mitogen-activated protein kinase (MEK)/extracellular signal-regulated protein kinase1/2 (ERK1/2) pathway [111]. By utilizing gain-of-function and loss-of-function approaches in vivo and in vitro, studies have further confirmed the critical role of the SR and cAMP signaling axis in developing DRs and fibrosis [47,108,112,113,114]. 

During the administration of CCl4, large cholangiocytes are selectively damaged over small cholangiocytes. Moreover, small cholangiocytes acquire the phenotype of large cholangiocytes during this kind of injury [35]. It has been determined that IP3/Ca^2+^ signaling regulates the proliferation of cholangiocytes differently. The proliferation of small cholangiocytes is specifically regulated by the α1-adrenergic receptor/Ca^2+^/calcineurin-dependent activation of the nuclear factor of activated T-cells (NFAT2) transcription factor and Sp1, which could function as a compensatory mechanism when larger cholangiocytes are significantly damaged [115]. Accordingly, other studies have successfully demonstrated the roles of cAMP and Ca^2+^ signaling in the transition of small cholangiocytes to large cholangiocytes when the function of the latter is compromised [33,116]. Additionally, β-adrenergic receptor signaling has also been demonstrated to promote DRs in a DDC model [117].

Infiltrated mast cells have also been reported to induce cholangiocyte injury in mouse models [118]. Mast cells are immune cells with a hematopoietic lineage, and they play a pivotal role in innate and adaptive immunity. Studies have demonstrated that mast-cell-derived TGF-β is a critical regulator of liver injury, and blocking its activity ameliorates liver injury [119]. Additionally, mast cells have been demonstrated to regulate FXR signaling and DRs in animal models [120]. The introduction of mast cells caused cholangiocyte injury, inflammation, and DRs in normal mice, suggesting its potential to initiate bile duct injury without any other external stimuli [121]. This study also demonstrated that mast cells preferentially interact with larger cholangiocytes via H2 histamine receptors and regulate their proliferation [121]. A recent study identified a specific population of DR-associated neutrophils that directly impacted the expansion of cholangiocytes in chronic liver diseases. This study further demonstrated that the depletion of neutrophils or the inhibition of their recruitment reduced DRs and disease progression [122]. In human BA biopsy livers, interleukin 8 (IL-8) was shown to be highly expressed in cholangiocytes, and there was a positive correlation between its expression level and bile duct proliferation [123]. Human cholangiocytes proliferate in response to interleukin 6 (IL-6), hepatocyte growth factor (HGF), and epithelial growth factor (EGF) in vitro [124,125]. Using a combination of in vitro and in vivo loss-of-function approaches, studies have demonstrated that the HGF/MET pathway preferentially induces the differentiation of LPCs towards a hepatocyte lineage via the AKT and STAT3 signaling axis, whereas the EGF/EGFR axis promotes the cholangiocyte differentiation of LPCs through a NOTCH1-dependent mechanism [126]. The elevated expression of various cytokines and their receptors was also reported in human BA, PSC, and PBC livers [127,128]. In a BDL model, increased expression levels of EGF, IL-6, basic fibroblast growth factor (bFGF), and TGF-β were reported, suggesting their role as positive regulators of DRs in vivo [129]. Additionally, other studies have demonstrated the role of estrogens and the vascular endothelial growth factor (VEGF) in cholangiocyte proliferation [130,131]. Prolyl 4-hydroxylase (P4HA2), a key enzyme involved in collagen synthesis, was positively correlated with DRs and fibrosis in human PBC and PSC livers. Furthermore, a *P4HA2*−/− *Mdr2*−/− double knockout mouse model displayed reduced levels of DRs and fibrosis compared to *Mdr2*−/− mice [132].

Macrophages are essential cell types known to influence liver function in different ways. In the absence of injury, healthy mice injected with bone marrow-derived cells developed DRs and demonstrated that donor macrophages were the source of the TNF-related weak inducer of apoptosis (TWEAK), a known cytokine that promotes cholangiocyte proliferation. The cholangiocytes in the recipient mice tested positive for its receptor fibroblast growth-factor-inducible protein 14 (Fn14), suggesting the presence of an active TWEAK/Fn14 pathway. The deletion of TWEAK/Fn14 signaling in macrophages reduced the DRs, further confirming the role of this pathway in DRs [133]. Moreover, the inhibition of the TWEAK/Fn14 pathway reduced DRs and fibrosis in an experimental model of BA [134]. In support of these findings, increased Fn14 expression was observed in human BA livers compared to normal controls. Additionally, stem/progenitor cell marker PROM1/CD133 expression has been demonstrated to play a role in developing DRs and fibrosis in BA livers [11,135,136]. 

Notch signaling is yet another signaling axis that regulates DRs. Notch signaling mediates cell proliferation, cell differentiation, and the development of various organs, including the liver [137,138,139]. In an experimental model of BA, activated Notch signaling induced DRs. Notably, dysregulated Notch signaling was reported in human BA and Alagille syndrome livers [140,141,142]. The lineage tracing studies demonstrated that Notch signaling mediated the reprogramming of hepatocytes to a cholangiocyte phenotype and contributed towards DRs in a DDC-diet-induced liver injury [49]. During biliary regeneration, the expression of the Jagged1 protein produced by the myofibroblasts resulted in the activation of Notch signaling in LPCs, promoting their differentiation towards cholangiocytes [143]. Notch signaling was also suggested to cause DRs downstream of cytokeratin 19 (CK19) signaling in mice subjected to a DDC diet [144]. In an experimental model of thioacetamide-induced liver injury, hepatocyte reprogramming was mediated via a COX2-TGF-β-TGFbR1-β-catenin-dependent mechanism [145]. The presence of activated TGF-β signaling in hepatocytes with a biliary phenotype in human biopsy livers supports the role of the TGF-β pathway in the cellular reprogramming process and its role in DRs [146]. The connective tissue growth factor (CTGF) is a matrix protein that mediates cell-to-matrix interactions through various subtypes of integrin receptors and induces TGF-β activation. A *Ctgf* deficiency or the inhibition of integrin αvβ6 reduced cholangiocyte proliferation in a DDC injury model [147].

Studies have demonstrated a functional interaction between β-catenin, CFTR, and NF-κB in cholangiocytes. A loss of either β-catenin or CFTR resulted in the activation of NF-kB, promoting inflammation and DRs [148]. NF-κB-inducing kinase (NIK) is primarily known for activating the noncanonical IKKα/NF-κB2 pathway, which regulates immune functions. Cholangiocyte-specific NIK increases cholangiocyte proliferation while mitigating cholangiocyte death. NIK also promotes the secretion of cytokines from cholangiocytes, which can have additional autocrine mitogenic effects and, thus, amplify the formation of DRs [149]. 

Obstructive cholangiopathies are characterized by the accumulation of toxic bile acids in the liver, which are known to cause cellular toxicity. Studies have shown that bile acids increase cholangiocyte proliferation and secretin-stimulated ductal secretions, both in vitro in cholangiocytes and in vivo in a bile-acid-fed experimental model [150,151]. Bile acids specifically induce the proliferation of large cholangiocytes but not small cholangiocytes [150]. Further studies have demonstrated that bile acids increase the expression of the Na^+^-dependent apical bile acid transporter, which enhances bile acid uptake and initiates cholangiocyte proliferation via the PI3K-AKT pathway [152]. 

The Hippo/YAP signaling pathway is another important regulator of cell growth and differentiation in the liver [153]. YAP activation in hepatocytes induces their reprogramming toward the biliary phenotype via Notch activation [154]. Studies have determined that YAP and mTORC1 signaling promote DRs [95]. In hepatocytes, YAP signaling is induced by bile acid and promotes hepatocyte reprogramming into biliary cells upon injury [155]. Furthermore, YAP knockout mice developed severe bile duct paucity and necrosis compared to controls, demonstrating that YAP signaling is necessary for cholangiocyte survival under normal conditions [155]. There is evidence that hepatocyte-to-cholangiocyte reprogramming is regulated via the Wnt/β-catenin signaling pathway. Mice overexpressing a stabilized form of β-catenin showed an increased number of cells that expressed markers of both hepatocytes and cholangiocytes after a DDC-diet-induced liver injury [156]. A recent lineage-tracing study on a zebrafish model also demonstrated that the Notch-YAP signaling pathway regulates hepatocyte reprogramming toward a cholangiocyte lineage [53].
cells-13-00579-t002_Table 2Table 2Major signaling pathways that regulate DRs.PathwayInjury ModelEffect on DRsReferencesNotchBDL, DDC, MCD↑[49,93,140]cAMP/PKA BDL↑[111,157]Secretin–secretin receptor BDL↑[112,113]YAPDDC, *Mdr2*−/−, CCl4↑[95,132,158]IL6R/HGFBDL↑[106]SerotoninBDL↓[107]5HT/5HTR2A/2B/2C axisBDL, *Mdr2*−/−↑[108]TGF-beta *Mdr2*−/−↑[119]FXR*Mdr2*−/−, DDC↑[120,159]EGFRDDC↑[126]COX2/TGF betaTAA↑[145]CNN1/α_v_β_3_/NfkB/NotchBDL↑[160]HedgehogBDL, CCl4↑[158,161]TWEAK/Fn14 pathway CDE, RRV↑[134,162]IL-33RRV↑[163]

Hedgehog (Hh) signaling is one of the evolutionarily conserved pathways that play an important role in liver development and morphogenesis [164]. Cholangiocytes express both Hedgehog (Hh) ligands and receptors [165]. In BDL and human PBC livers, cholangiocytes were positive for Hh ligands, receptors, and their target genes [161,166]. Hh signaling is activated in human BA and is primarily localized around the biliary ducts, possibly accounting for its role in DRs [167]. Another study determined that activated Hh signaling in peribiliary myofibroblast cells contributed to the progression of DRs, fibrosis, and peribiliary remodeling in vivo [168]. Studies have also demonstrated the epigenetic regulation of DRs in animal models. Mice with a cholangiocyte-specific knockdown of p300, one of the histone acetyltransferases, developed fewer DRs after BDL than wild-type mice. It has also been determined that the long noncoding RNA ACTA2-AS1 recruits the p300/ELK1 transcriptional complex to specific gene promoters, which regulates the expression of pro-ductular and pro-fibrogenic genes in this model [169]. A summary of the well-studied signaling mechanisms associated with DRs and cellular transdifferentiation during chronic injury is presented in Table 2.

## 6. Translational Significance of DRs

Pathological DRs in human liver diseases are associated with chronic inflammation, progressive fibrosis, and LPC expansion, which have been successfully recreated in many experimental models. Chronic liver diseases caused by either an infection, toxic substances, metabolic errors, or genetic factors are all associated with cellular damage and cell death. Injured and apoptotic cells serve as the sources of stimuli to initiate the replication of existing cells to replace the dying cells. Cell proliferation under normal conditions is tightly regulated via intricate signaling networks. DRs result from the imbalance between cell death and proliferation during injury. Therefore, the timely regulation of cholangiocyte/hepatocyte apoptosis and proliferation should be maintained to achieve tissue homeostasis during chronic injury.

Several high-impact studies have shown that DRs consist of transitional cell types that express markers of cholangiocytes, hepatocytes, mesenchymal cells, myofibroblasts, immune cells, and progenitors in chronically injured livers. This intrinsic ability to express multipotent genes is possibly a conserved survival mechanism in rare populations of mature liver cells under chronic conditions. These cells could have originated from the same precursor or stem cells during embryonic development [170]. Although the link between DRs and the progression of liver diseases is indisputable, pathological DRs are strongly correlated with fibrosis progression and disease severity in many chronic liver diseases. A transcriptomic analysis at the single-cell level revealed a significant heterogeneity with novel transitional cell types, highlighting the spatial complexity at molecular and cellular levels in DRs [171]. 

Even though experimental models show the regenerative capacity of hepatocyte-derived cholangiocytes and LPCs, more in-depth molecular studies are required to determine the translational significance of reprogrammed hepatocytes and LPCs in chronic liver diseases. Lineage-tracing studies conducted on mice showed that reprogrammed hepatocytes could lead to the formation of cholangiocarcinoma and hepatocellular carcinoma [172]. The cellular transformation potentially occurs due to epigenetic changes and the dysregulation of signaling pathways activated during injury-associated cellular reprogramming. When exposed to persistent and chronic injuries, activated cholangiocytes secrete various chemokines and cytokines that affect other cell types, which, in later stages, progress into fibrosis, cirrhosis, neoplastic transformations, and tumorigenesis [173,174]. Cholestatic liver diseases, such as PSC and PBC, are known to present a higher risk of developing cholangiocarcinoma. Nearly 10% of PSC patients are known to develop cholangiocarcinoma [13,175,176,177]. In metabolic diseases, DRs could be an even more complicated cellular response promoting disease progression. The EVs derived from DRs and LPCs may serve as a systemic source of inflammatory and procoagulant factors, which could increase the cardiovascular disease risk [178,179]. Therefore, further studies are needed to fully understand the cell lineages and translational significance of transitional cell types, including reactive cholangiocytes, hepatocyte-derived cholangiocytes, and LPCs, in chronic liver diseases.

The presence of invasive DRs consisting of LPCs and hepatocyte-derived cholangiocytes in diseased human liver biopsies and chronic hepatic injury models led to the hypothesis that DRs promote tissue regeneration. These newly formed ductular structures with bile canaliculi likely serve as bile-draining channels during chronic injury. Additionally, an increase in the cholangiocyte mass with no definite lumen may contribute to hepatic functions such as xenobiotic, drug, and cholesterol metabolism. In these cases, DRs may also contribute to the repair during chronic liver injury by a subset of cholangiocytes that express hepatocyte markers. This outcome was observed in lineage-tracing studies in in vivo mouse models [60,180,181]. Importantly, in human liver disease biopsies presenting with invasive DRs, the novo hepatobiliary canalicular junctions were evident and could be a protective mechanism by enhancing bile drainage [182]. Thus, DRs could be a reparative response in a subset of pathologies where tissue homeostasis is attained within a definite period of time. 

Although the contribution of LPCs to liver repair has been determined in several experimental models, the influence of an altered microenvironment on LPCs in chronic injury and their role in disease progression in chronic liver diseases remain elusive. Their close association with fibrosis progression and the expression of multilineage markers, including proinflammatory and profibrogenic phenotypes, must be investigated critically to determine their role in repair versus disease progression [11,67,136]. An interesting study conducted by Planas-Pas et al. followed a single-cell analysis of EpCAM-positive cholangiocytes isolated from DDC-injured mouse livers and identified three distinct clusters of cholangiocytes, consisting of actively proliferating cholangiocytes, proinflammatory cholangiocytes, and hybrid cholangiocytes expressing hepatocyte markers. The authors concluded that proliferating cholangiocyte cells likely contributed to DRs, while proinflammatory cholangiocytes induced inflammation and the activation of hepatic stellate cells, potentially promoting fibrosis and hybrid cholangiocytes involved in liver regeneration [95]. Characterizing these cells on a molecular level will provide a new perspective on future therapeutic strategies to enhance the pro-regenerative potential of cholangiocytes and LPCs while selectively targeting proinflammatory signals to ameliorate the progression and severity of chronic liver diseases. 

## 7. Conclusions

The outcome of DRs is multifactorial and largely depends on the nature of the etiology and other complex cellular and molecular events that occur in response to an injury. Exploiting the reparative mechanisms while targeting profibrogenic/inflammatory/oncogenic pathways at the critical phase of liver injury may enhance the regenerative ability of liver cells while suppressing the detrimental effect of DRs in chronic liver diseases. Elucidating the dysregulated signaling pathways that promote cellular reprogramming and transdifferentiation may provide a better understanding of the compromised regeneration mechanisms in chronic liver diseases. Cell-to-cell communication plays a pivotal role in developing diseases such as liver fibrosis. Recent studies have demonstrated that EVs play a significant role in the progression of various liver diseases [183]. EVs are known to be released from reactive cholangiocytes, LPCs, and senescent hepatocytes. A detailed investigation of DR-derived EVs could lead to the discovery of novel disease-specific biomarkers and their therapeutic potential in chronic liver diseases. Future studies focusing on elucidating the complex signaling mechanisms associated with DRs and how they influence the microenvironment and regulate the tissue-remodeling process during disease progression at the single-cell level by adopting advanced spatial biology technologies and omics approaches will enhance our current knowledge of DRs and their unidentified pathophysiological role in chronic liver diseases. These discoveries will direct future research efforts to develop novel cell-type targeted therapies to manage liver fibrosis and other chronic liver diseases associated with a pathological DR.

## Figures and Tables

**Figure 1 cells-13-00579-f001:**
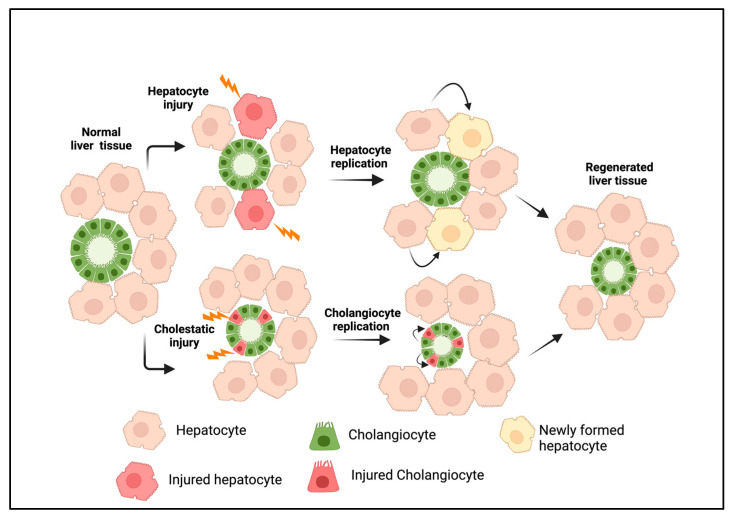
Liver regeneration during acute injury. Hepatocyte injury or loss induces the replication of existing hepatocytes, whereas cholangiocyte injury promotes the proliferation of cholangiocytes to replace injured cells. This results in tissue homeostasis under a normal physiological state. The figure was created using BioRender (https://biorender.com).

**Figure 2 cells-13-00579-f002:**
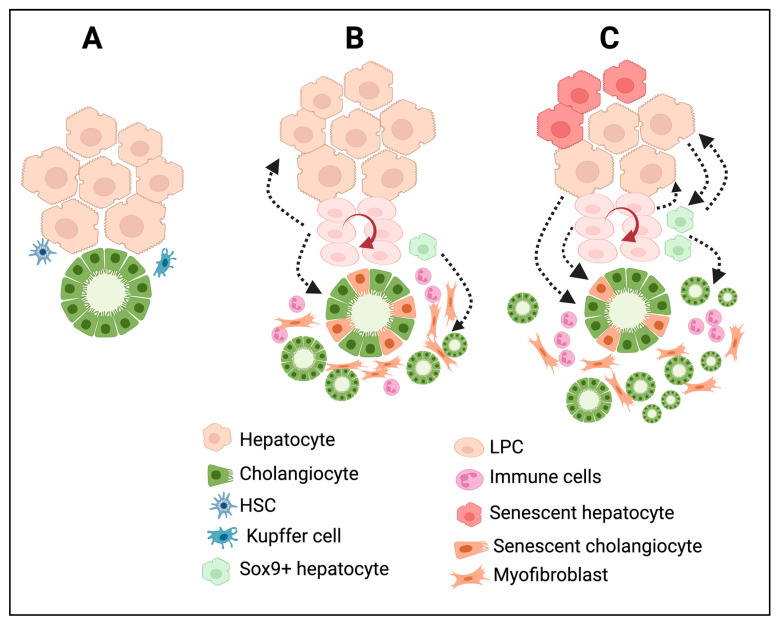
DRs are associated with extensive cellular reprogramming and tissue remodeling during chronic liver injury and diseases. (**A**) Bile duct and liver parenchymal architecture under normal physiological conditions. The peribiliary area is surrounded by quiescent liver parenchyma and mesenchymal cells such as HSC and Kupffer cells. (**B**) In chronic cholestatic liver injury; the damaged cholangiocytes trigger immune infiltration, cholangiocyte proliferation, differentiation of periportal SOX9^+^ hybrid hepatocytes towards cholangiocyte phenotype, activation of LPCs and its differentiation, activation of HSC towards myofibroblast cells resulting in the formation of invasive DRs and fibrosis. (**C**) Chronic hepatic insult results in extensive reprogramming of hepatocytes and cholangiocytes, activation and differentiation of LPCs, immune infiltration, and formation of several intermediate cell types expressing markers of hepatocytes, cholangiocytes, LPCs, and myofibroblast cells at various degrees, resulting in a distorted tissue architecture in the liver with invasive DRs and fibrosis. The figure was created using BioRender https://biorender.com.

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
