# Peer review of "Ductular Reactions in Liver Injury, Regeneration, and Disease Progression—An Overview"

_cells, 2024, doi:10.3390/cells13070579_

Round 1
Reviewer 1 Report
Comments and Suggestions for Authors
Mavila et al. illustrate with precision the complex picture of ductular reaction in chronic liver disease. They provide a detailed summary of all the aspects of this phenomenon; its etiology, its evolution during the injury, cell types involved and the molecular mechanism underlying. Nevertheless, there are some issues to be addressed:
In lines 64-70, the authors enumerate the morphological differences between cholangyocite populations, how are these morphological differences related to functional differences?
In lines 79-80, authors affirm that cholangyocites secrete immunoglobulins, but there is no reference to support this statement. Which immunoglobulins are secreted?
In lines 149-151, the authors comment about signals regulating the microenvironment of CoH. In this point, I suggest going into a little mor detail about these signals that are fundamental to LPC activation and therefore to DR.
In line 166, it is stated that LPCs “contributed towards inflammation in chronic liver damage”. How they do it? Attracting neutrophils, activating macrophages or lymphocytes…?
Figure 2 needs to be corrected. Legends labels do not match with symbols, which leads to confusion.
In line 266-269, effects of HGF and EGF in human cholangiocytes are described. However, these growth factors also modulate the biology of LPCs. A brief mention of this could be interesting.
At the end of point 4, a summary table of the effects of each pathway mentioned could be useful.
MINOR POINTS
In figure 1 there are presented purple cells which are not mentioned in the legend or in figure caption.
In line 68, after “surface”, is missing a dot.
In line 128, the reference to the studies mentioned above is missing.
In line 158, appears CoA instead of CoH.
Reviewer 2 Report
Comments and Suggestions for Authors
This is a review article of ductular reaction (DR) in liver diseases. This topic could be attractive for pathologists and researchers interested in many liver diseases. DR is a common finding in many acute and chronic liver disorders since was first described on 1956 by Emmanuel Farber at the University of Toronto when he injured rat livers by partial hepatectomy with a compound (2-acetamynofluorene) to block hepatocyte proliferation. In the same experiments, Farber also coined the term oval cell which were sometimes related to stem cells of the liver. Although it became clear from different injury models in rodents that activated oval cells was different depending on the nature of the liver these populations were all encompassed by the term Liver progenitor cell (LPC) or as in this manuscript Hepatic Progenitor Cells (HPC). Similarly, both DR and HPC were described in many human chronic diseases like biliary atresia (BA), primary sclerosing cholangitis and so on. Following the liver insult, HPC almost always emerge from the periportal area termed the canal of Hering which are situated between the bile canaliculus and the bile ducts. However, more recent lineage tracing experiments in the mouse liver have found many contradictory data of the existence of stem cells in the adult liver. For example, Grompe et al [1] showed that the only significant origin for replacement hepatocytes is through proliferation themselves being less than 1% coming from clonogenic progenitors derived from biliary system. Therefore, taking into account these considerable contradictory and controversial data that remain for the future and some flaws in exposing these data some concerns with this manuscript are raised:
- A real definition of DR is lacking. Probably DR should be defined as “reaction of ductular phenotype, possibly but not necessarily of ductular origin according to Roskams[2]. DR is histologically observed in liver specimens and pathologically recognized as bile duct proliferation or hyperplasia. I would suggest the author to start with a correct definition of the term
- The title of the article could be improved. Given duct DR is commonly found in many liver diseases and is described in the manuscript also after some kind of acute injuries to the liver the title should not be restricted to “chronic liver injury and fibrosis”. Other revisions actually do in that way[2] . In fact, in line 395 it is stated “ DR may be reparative response in a subset of pathologies where injury is acute..”
- Some statements are not correctly referenced. For instance, in line 27 no reference was found. I would suggest to review all the article to support every sentence by appropriate and recent bibliography.
- Given the many exposed data are contradictory in the actual literature (as previously shown) some “tone down” in several statements are to be more appropriate. For instance, in line 147 it was stated that LPC has been demonstrated in human livers but it should be clarified that it was usually not found in normal livers.
- Some descriptions in the manuscript are not to the point of DR and may be useless. For instance, in line 220 a long paragraph about gene expression including secretin receptors is included. These descriptions could be omitted or at least reduced because it is outside the scope of the manuscript.
- Figure 2 is misleading. In the legend some errors are detected. Por instance, the picture for normal and injured hepatocyte is the same. Furthermore, in liver at normal stage there should not be activated HSC as it is shown. Please revise.
- Some spelling errors are to be corrected: line 68 (“surface Besides”) or in line 266 (“choalngiocytes”)
BIBLIOGRAPHY
[1] B. D. Tarlow et al., “Bipotential adult liver progenitors are derived from chronically injured mature hepatocytes,” Cell Stem Cell, vol. 15, no. 5, pp. 605–618, 2014.
[2] K. Sato, M. Marzioni, F. Meng, H. Francis, S. Glaser, and G. Alpini, “Ductular Reaction in Liver Diseases: Pathological Mechanisms and Translational Significances,” Hepatology, vol. 69, no. 1, pp. 420–430, 2019.
Reviewer 3 Report
Comments and Suggestions for Authors
The manuscript by Mavila and colleagues describes in depth the types of ductular reaction (DR), their cellular derivation, and the biological and pathophysiological significance of DR. Furthermore, Authors highlighted and described the mechanisms modulated by DRs in driving the aberrant reparative/regenerative response found in cholangiopathies. The manuscript is well written, clear and the illustrations supporting the text are well done and well explanatory. Despite appreciation for the manuscript, some points could be better explored, some references added, and, finally, some figures/tables could be added to increase the clarity of the text.
In the introduction and in the subsection concerning the subsection concerning the role of the primary cilium of cholangiocytes, in addition to the seminal works of the "Leuven school" and the Mayo clinic, a very informative and well-written review by Banales and colleagues from 2019 should be cited (PMID : 30850822).
In the chapter regarding the derivation of cholangiocytes from hepatocytes, the hypothesis was also explored and demonstrated by Calvisi and Chen, using transfection models with hydrodynamic tail vein injection and also, in a recent work, (PMID: 36626626) on Zebrafish. A comment in this regard should be added to the text.
In lines 200-201, the association between DR and metabolic syndrome is mentioned. A recent review (PMID: 37951510) describing the DR-metabolic syndrome link and the growing interest in this field could be briefly commented.
The importance of the Notch signaling pathway in modulating the correct tubulization of biliary structures has been illustrated in Alagille syndrome and biliary atresia also by works by Libbrecht and Fabris (PMID: 15897750 and PMID: 17600123, respectively), which are worthy of mention.
The part of the manuscript dealing with modulation of serotonin receptors should be implemented with further work by Kyritsi and colleagues (PMID: 31344280). Furthermore, a comprehensive review of the morphogens involved in the development of biliary structures was proposed by Strazzabosco (PMID: 22245898) that could be cited.
Given the complexity of the mechanisms described in chapter 4, a figure summarizing the main molecular mechanisms mediated by DR could be added to help non specialists in cholangiopathies.
A recent review by Banales and ENS-CCA (PMID: 32606456) should be added to lines 379-380, where the association between DR and neoplastic transformation is discussed.
For completeness, it might be interesting to add a brief chapter regarding models to study DR, liver fibrosis and cholestasis (https://doi.org/10.1016/B978-0-12-415894-8.00015-4, PMID: 30398152, PMID: 33486884).
The conclusions should be expanded and made more personal, with an excursus also to the therapeutic possibilities that could be opened up by targeting DRs.
There are some typos and some spaces between words have been doubled, please correct
The style and correctness of the English language is optimal
Round 2
Reviewer 3 Report
Comments and Suggestions for Authors
The quality of the manuscript has now significantly improved and the criticisms raised by the reviewer have been addressed